# The VIPR-1 trial (Visualizing Ischemia in the Pancreatic Remnant): Assessing the role of intraoperative indocyanine green perfusion in predicting postoperative pancreatic leaks and fistulas: Protocol for a phase II clinical trial

**Gustavo Salgado-Garza**[1], **Annika Willy**[1], **Flavio G. Rocha**[1,2,3], **Skye C. Mayo**[1,2,3], **Brett C. Sheppard**[1,2,3], **Patrick J. Worth**[1,2,3]*

1 Oregon Health & Science University, Department of Surgery, Portland, Oregon, United States of America, 2 Brenden Colson Center for Pancreatic Care, Oregon Health & Science University, Portland, Oregon, United States of America, 3 Knight Cancer Institute, Oregon Health & Science University, Portland, Oregon, United States of America

* worth@ohsu.edu

## Abstract

Surgery of the pancreas has come a long way since its inception; however, postoperative morbidity is still high. Pancreatic leaks and fistulas are common complications in patients undergoing surgery to remove the pancreas. Fistulas delay subsequent oncological care after surgery and prolong the hospital stay. Hypoperfusion of the pancreas has been proposed as one factor leading to fistulas. Indocyanine green (ICG) injection allows the surgeon to evaluate blood perfusion to tissue in real-time. This protocol describes a trial that aims to assess the effectiveness of intraoperative ICG metrics of the cut edge of the remnant pancreas to predict postoperative fistulas. A single group will participate in an observational, surgeon-blinded, phase II trial. ICG measurements of the cut edge of the pancreas will be recorded before reconstruction. International Study Group on Pancreatic Surgery criteria for pancreatic fistula will be used to define leaks and fistulas. The study objective is to analyze the correlation between ICG measurements and the development or absence of both biochemical leak and clinically relevant fistula formation. Currently, limited objective intraoperative predictors exist for predicting postoperative fistulas. Having a reliable predictive tool could decrease the healthcare burden posed by fistulas. The findings of this trial will provide conclusions on the usefulness of ICG measurements in predicting postoperative pancreatic fistulas and leaks. This clinical trial is registered in ClinicalTrials.gov with the ID NCT06084013. The current protocol version is v1.1.

**Data availability statement:** No datasets were generated or analysed during the current study. All relevant data from this study will be made available upon study completion.

**Funding:** This study is supported by a Medical Research Foundation of Oregon grant. The funders did not participate in the study design and will not be involved in data collection, analysis, or dissemination of results.

**Competing interests:** The authors have declared that no competing interests exist.

## Introduction

Pancreatic cancer is a deadly disease with a growing incidence rate, causing over 50,000 deaths annually in the United States [1]. The 5-year survival rate for this cancer is very low, but it can increase to over 30% with successful surgical resection [2]. However, one of the most serious complications of pancreatic surgery is the development of postoperative pancreatic fistulas (POPF) or leaks, which can occur in up to a third of patients [3]. The impact of POPF on patients is twofold; it not only increases immediate postoperative complications but also adversely affects medium to long-term oncological outcomes. POPF increases postoperative mortality because of prolonged hospital stays and higher reoperation rates [4]. In addition, POPF is also linked to worse oncological outcomes for patients, notably tumor recurrence and overall survival [5,6].

Current fistula risk prediction models do not include modifiable intraoperative risk factors for mitigating pancreatic fistulas [7]. The surgical neck of the pancreas has a variable blood supply and is vulnerable to ischemia, a known key factor in developing leaks [8,9]. Indocyanine green (ICG) fluorescence is used to assess tissue perfusion during surgery, but its effectiveness in predicting pancreatic leaks has yet to be systematically studied. A reliable intraoperative method to analyze blood perfusion could help guide surgical margins and reduce the risk of pancreaticojejunostomy leaks, ultimately improving patient outcomes.

Indocyanine green (ICG) is an FDA-approved dye that emits fluorescence when near-infrared light is applied, making it useful for visualizing tissue perfusion during surgery. ICG is injected intravenously, traveling to tissues via arteries. While ICG is now considered standard in some surgical disciplines, it is not the case in pancreatic surgery, primarily due to a lack of adequately powered prospective studies analyzing the benefits of its use and surgeons wanting more evidence to change their practices [10]. Regarding pancreatic surgery, ICG is mostly used for anatomical identification of structures to improve surgical safety, avoid damage to critical structures, and delineate tumors for excision [11–14] Pancreatic surgeons have shown interest in further studies on ICG for better visualization of tissue perfusion. A recently published Delphi consensus study reported that almost 80% of experts agreed on the need to focus research efforts on ICG in predicting the risk for POPF [10]. Although research on ICG for evaluating leaks after pancreatectomy is limited, one study suggests its potential for detecting leaks in a single patient retrospectively [13]. Moreover, ICG has an excellent safety profile, with Clavien-Dindo grade ≥4 complications occurring in only 0.05% of cases where it is utilized [15].

This study protocol aims to investigate the relationship between hypoperfusion during surgery and the development of postoperative leaks.

## Materials and methods

This is a protocol for a Phase II, open-label, surgeon-blinded, observational study to assess the predictive potential of intraoperative perfusion parameters at the pancreatic cut edge via ICG and its relation to fistula formation versus conventional measurements. Conventional measurements would include subjective descriptions of known risk factors for fistulas and leaks such as pancreatic gland texture and

pancreatic duct size caliber. The study design consists of a single-arm intervention group at a single academic center: the Oregon Health & Science University Hospital in the United States. Following international and national definitions, this center is considered a high-volume hospital for pancreatectomies [16,17].

This study protocol adheres to the SPIRIT guidelines for trial reporting [18]. Our full SPIRIT Checklist can be seen in S1 File. To ensure adequate representation of ethnic and racial group demographics in the study, a full consent form and protocol will be available for English- and Spanish-speaking participants. Short summaries of the consent forms are available in other languages. Written consent will be obtained from participants by members of the surgical department, including clinical staff and research assistants. This project underwent full ethics review by the Oregon Health & Science University's Institutional Review Board with 25055 as a unique identificatory code. More details of the administrative and IRB approvals can be seen in S2 File. This clinical trial is registered in ClinicalTrials.gov with the ID NCT06084013.

## Sample size

Given the limited data of ICG perfusion in the pancreas, the sample size calculation was based on determining a detectable difference between the normal and poor perfusion groups. We anticipate a 20% incidence of fistula in the normal perfusion group, and at least a 65% incidence in the poor perfusion group is expected. With an expected imbalance ratio of approximately 4:1 between these groups, a total sample size of 53 participants (42 in the normal perfusion group and 11 in the poor perfusion group) provides 80% power at a 5% significance level to detect this difference. To account for potential exclusions due to postoperative factors associated with leaks, we anticipate screening 75 participants and enrolling 50.

## Eligibility criteria

All participants must fulfill eligibility criteria and have none of the excluding criteria.

Inclusion criteria:

1. Participant scheduled for elective pancreaticoduodenectomy for any diagnosis.

2. Participants≥ 18 years of age.

3. Ability to understand the nature and individual consequences of clinical trials.

4. Written informed consent from the participant or legally authorized representative.

5. For participants of childbearing potential, a negative pregnancy test and adequate contraception until 14 days after the trial intervention.

6. Participant needs to have an operative drain (any closed suction drain) after the procedure.

7. Participants that do not require arterial reconstruction.

8. Participants that require minor portal venous reconstruction including patch venoplasty.

Exclusion criteria:

1. Patients with previous history of adverse reaction to contrast dye, ICG or components of the dye.

2. Prior pancreatectomy.

3. Known diagnosis of hepatic insufficiency, hepatitis, liver fibrosis or cirrhosis, or chronic pancreatitis.

4. Because this study focuses on hypoperfusion, patients will be excluded if in postoperative day 3–5 had any of the following: persistent SBP<90 mmHg unresponsive to 1L crystalloid, unexpected ICU transfer, blood transfusion of >2 units intraoperatively or 1 postoperatively, continuous vasopressor treatment or ACLS protocol initiation.

5. Organ failure, anuria or NSQIP-identified complication will be reviewed by PI and attending surgeon and decide exclusion.

6. Patients who require arterial reconstruction as part of their procedures.

## Blinding

To avoid the influence of ICG measurements on surgical decisions at this stage of the trial, this study will involve surgeon-blinding. During intraoperative measurements, the primary surgeon must either step outside the operating room or be positioned to avoid a line of sight to the imaging device monitor. The secondary surgeon will remain scrubbed in for this portion of the case. Both study personnel and operating room staff will ensure this blinding process is successful. Any instances of the surgeon viewing the measurements will result in the participant's removal from the study. To ensure adequate surgeon-blinding, objective measurements of the captured images are performed postoperatively, with no relay of this information to the surgeons during the 30 days of the active study phase. Since this study is open-label, participants will know if they received the dye and underwent intraoperative measurements. As this intervention is not standard of care, there is no need for premature unblinding to the surgeons. Imaging measurements will only be available to surgeons and participants after the end of the study follow-up, which is 30 days after surgery.

## Intervention

During pancreatectomy, after the surgical specimen is removed and before creating the pancreatojejunostomy anastomosis, a single dose of 12.5 mg (5mL once reconstituted) of ICG will be administered intravenously by the anesthesiologist or certified nurse anesthetist. The Spy+ Elite near-infrared imaging platform (Stryker, USA) will be used to record perfusion metrics, taking as reference the gastric body and small intestine to compare differences in perfusion of the pancreatic stump. Recording of a video image starts as soon as the dye is injected. Still pictures will be extracted from the video at 10-second intervals once tissue saturation is stable, allowing a comparison of perfusion to the pancreatic stump and the reference tissue. The video footage of intake and initial washout of ICG will be captured for at least 90 seconds per participant.

While measurements and recording of perfusion are being obtained, the primary surgeon will step out of the operating room to ensure that ICG-derived data does not influence the surgical plan. Upon the study's completion, descriptive statistics will be reported to analyze the impact of ICG use and perfusion measurements on pancreatic surgeries. During ICG administration, vital signs will be continuously monitored, and hypersensitivity symptoms will be managed promptly. An intra-abdominal drainage tube will be placed to assess for amylase levels.

Participants can withdraw consent at any time. Adverse effects from the study interventions will be monitored and reported to institutional authorities. Fig 1 shows the schedule for participants.

## Outcomes

**Intraoperative.** During the surgery, a 90–120-second video will be captured to analyze ICG perfusion metrics. Because no current gold-standard metric exists for this imaging technique in the pancreas, several metrics will be analyzed. Quantitative ICG metrics will include absolute and relative ROI scores within the Cinevaq (Stryker, USA) software, T0 (the time from ICG injection until the first fluorescent signal), TMax (time from the first ICG signal until maximum uptake is achieved), as well as ingress and egress rates for the contrast [19,20].

**Postoperative.** After the surgery, the participant would enter the study's follow-up period. The main outcome will be the development or absence of biochemical leaks or postoperative pancreatic fistula, as defined by the most updated definition of the International Study Group in Pancreatic Surgery (ISGPS) [21]. Following this, we will collect surgical drain amylase levels at day three after surgery for every participant, clinical conditions of the participant, imaging results, and persistent drainage from the drain, among other factors delineated in the ISGPS. Additional surgical drain amylase

| | STUDY PERIOD | | | | | |
|---|---|---|---|---|---|---|
| | Screening | Post-intervention (days) | | | | End of follow up |
| TIMEPOINT (days) | -1 | 0 | D₁ | D₂₋₇ | D₇₋₂₉ | D₃₀ |
| **ENROLLMENT:** | X | | | | | |
| **Eligibility screen** | X | | | | | |
| **Informed consent** | X | | | | | |
| **INTERVENTIONS:** | | | | | | |
| *Routine Pancreatectomy* | | | X | | | |
| *Intraoperative imaging* | | | X | | | |
| **ASSESSMENTS:** | | | | | | |
| *Health status, medical history, allergies* | | X | | | | |
| *Tumor size, pathology results* | | X | | | X | |
| *Drain amylase* | | | | X | | |
| *Signs and symptoms for fistula* | | | X | X | X | |
| *Postoperative imaging if fistula suspicion* | | | X | X | X | |
| *Primary outcome final assessment* | | | | | | X |

**Fig 1. Schedule of enrollment interventions and assessments.**

measurements that are done for patient specific care would be also recorded and reported. Fig 2 depicts a flowchart for individual participants in the trial.

Descriptive statistics will be used to compare patients that developed postoperative fistula. ICG metrics will then be compared with multivariable modeling to study the impact they might have on any grade postoperative fistulas/leaks. An analysis will be done to evaluate clinically relevant postoperative pancreatic fistulas (grades B and C per ISGPS) and biochemical leaks separately.

Because there are characteristics that might influence development of POPF, the following participant and case characteristics will be recorded and analyzed as covariates affecting outcomes: 1) type of pancreaticojejunostomy anastomosic technique (duct-to-mucosa, invagination, or use of duct stent), 2) principal surgeon, 3) operative length, 4) blood loss in mL, 5) intraoperative blood transfusion, 6) duct size in mm, 7) gland texture – soft, intermediate, firm, 8) intraoperative use of vasopressors, 9) race, 10) BMI, 11) age, 12) neoadjuvant therapy and 13) vascular reconstruction [22].

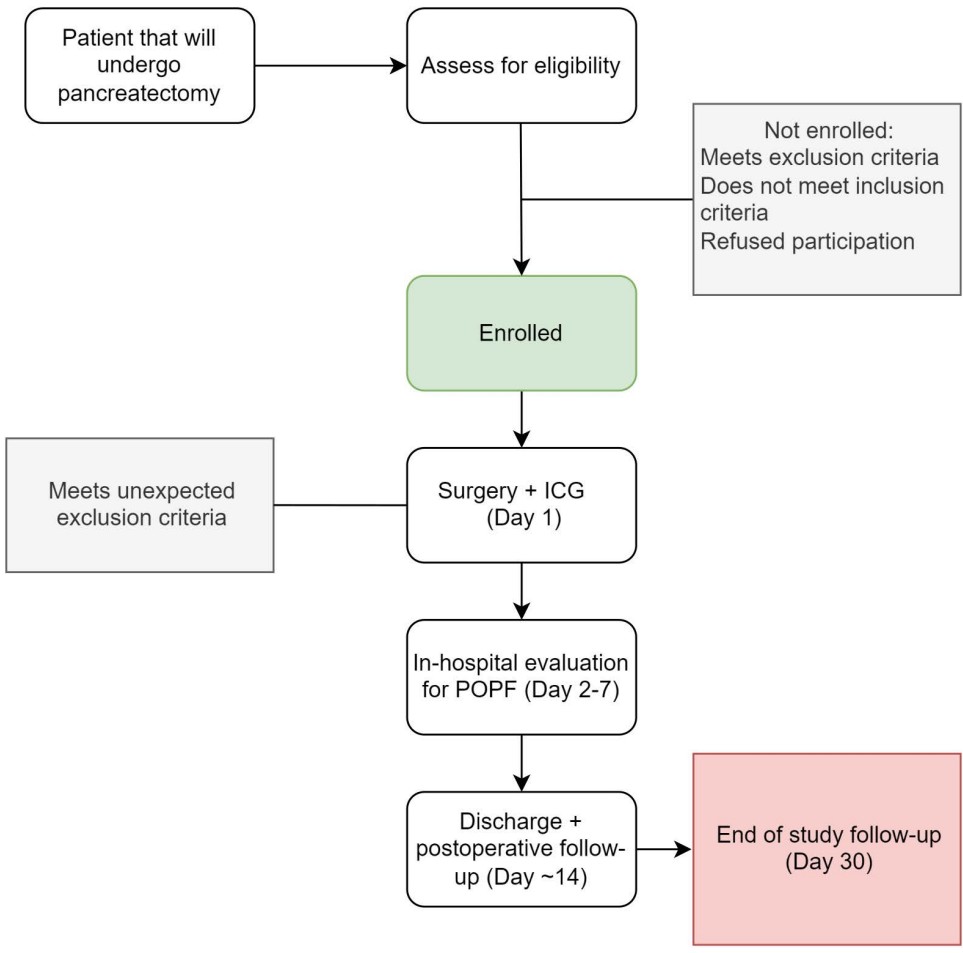

**Fig 2. Trial flowchart.** Flowchart for participants in the VIPR-1 trial.

## Data management

Data to be collected includes baseline participant demographics relevant to our study, including sex, age, ethnicity, race, height, weight, ECOG performance status, disease stage at study entry, cancer therapy and biomarker status.

Collected data will be stored in a trial-specific database that lives in an encrypted cloud service that is institutionally compliant. Here, participant-level information will be captured without any of the 18 HIPAA identifiers; a participant key will be stored separately and securely. This study will comply with the Data and Safety Monitoring Plan established by the Knight Clinical Research Quality & Administration.

## Safety considerations

There are two relevant safety considerations for this study. The first involves safety during the imaging procedure, and the second involves securing participant data. ICG is considered a relatively inert contrast agent compared to other more commonly used imaging agents. The most frequently encountered adverse effects of ICG injection include allergic-type reactions, presenting as nausea and urticaria, with life-threatening complications such as anaphylactic shock being rare (<0.05%) [23]. Participants are made aware of the potential consequences of ICG injection at the time of screening and consent.

All adverse effects are reported within 24 hours through the Clinical Trials Management System, connected to ClinicalTrials.gov. The Knight Cancer Institute, as the trial sponsors, performs constant auditing on a participant-level basis. Participant information will be kept secure by restricting access to direct study personnel, and all images and captured participant-level data will be kept in encrypted platforms.

### Ethical considerations

The VIPR-1 trial follows guidelines set out by the Declaration of Helsinki [24]. The trial underwent review and subsequent approval from the Knight Cancer Research Institute via the Clinical Research and Review Committee on 9/2023 and then the Institutional Review Board from Oregon Health and Science University on 11/2023.

### Status and timeline

The VIPR-1 trial is registered in the ClinicalTrials government platform [25]. Participant recruitment started on 06/10/2024, and the estimated finishing time is one year after the first patient enrollment. Any updates to the protocol will be informed in the trial registry platform and the protocol itself. The complete internal study protocol is provided in S3 File. After the trial is completed, we intend to publish our results in a scientific journal.

## Discussion

This is the first prospective clinical trial in the United States to evaluate pancreatic remnant perfusion using ICG to predict fistulas and leaks. A recently published systematic review compiled all reports of the use of pancreatic remnant perfusion intraoperatively. A total of 5 studies were found, and only two were prospective [26]. The first prospective trial used ICG for postoperative acute pancreatitis rather than POPF as a primary outcome measure, where adequate statistical power for POPF was not mentioned [14]. The other prospective trial looked at subjective descriptions of adequate blood flow to the pancreatic remnant to characterize hypoperfusion and guide margins [27]. Thus, further highlighting the need for prospective intraoperative and subjective evaluation of perfusion. While studies have reported that hypoperfusion detected by ICG is correlated with POPF, currently, no cutoff for considering adequate or inadequate perfusion exists. After operative measurements are performed, we intend to utilize the Cinevaq software (Stryker, USA) to measure the perfusion difference with reference to gastrointestinal tissue and within the pancreas. If a correlation between perfusion and leaks is detected, we will perform statistical analysis at different thresholds for poor perfusion to identify the best cutoff delta in perfusion to predict POPF. A recent study demonstrated that cutoff values for hypoperfusion in breast tissue do not translate between different imaging systems. Specifically, a 33% relative perfusion measurement has been validated in breast tissue, specifically in the SPY-Elite® platform. However, on the newer SPY-PHI platform, the same cutoff value did not correlate with hypoperfusion [28]. While studies like this are not available in pancreatic tissue, it is possible that a specific cutoff would need to be validated for each type of device model in pancreatic perfusion dynamics.

If the results are positive, a phase III study can be designed to evaluate whether ICG metrics can guide surgical margins to optimize perfusion of the pancreatic remnant and decrease POPF. The level of the transection of the neck of the pancreas has been previously characterized as important for POPF risk [29]. Capturing patient preoperative and intraoperative variables is crucial to adjust for possible confounders in perfusion observations and the development of POPF. All patients undergoing open pancreatectomy for any diagnosis will be screened for this study. Broad inclusion and exclusion criteria will be implemented to ensure a diverse group of patients and reach the sample size needed.

In the past decade, minimally invasive pancreatectomy has gained major interest, as evidenced by numerous studies examining ways to reduce postoperative complications. Current data are inconclusive regarding the impact of surgical technique (open versus minimally invasive) on POPF rates, with some studies suggesting benefits. In contrast, others indicate an increased risk of this outcome with minimally invasive surgery [16,30]. A recent proof of concept prospective study investigated the predictive potential of ICG in determining suture tension-induced hypoperfusion in pancreatic

reconstruction. In this study, hypoperfusion to the pancreas by ICG was more frequently encountered in those patients who developed fistulas [31]. This study, however, was performed only in robotic cases, and all measurements were taken after the surgeon placed sutures for the anastomosis. However, these study results were considered in our sample size calculations. Contrasted with this study, we intend to measure perfusion before the pancreas is sutured to the jejunum to measure the complete surface of the remnant to be anastomosed.

Pancreatectomies are among the most challenging and lengthy procedures in gastrointestinal surgery. Because of this, the trial's impact on time added to the case should be minimal. Increases in operative time elevate costs and staff utilization and are associated with surgical complications in pancreatic surgery [32]. To address this, the operative time will be compared to control cases that did not undergo ICG measurements. This comparison will determine the impact of ICG measurements on total operative time. Studies in other gastrointestinal surgeries using ICG have shown a potential increase in operative time by 10–15 minutes, but this increase has not been statistically significant [33]. We currently do not know the impact of ICG in open pancreatectomies. ICG is now standard in many gastrointestinal procedures but not for the pancreas. We intend to record and subsequently report the effect of ICG on operative time for pancreatectomies.

The main limitation of this study is that our findings might not be generalizable to other institutions because of inherent differences in practice standards, surgeon experience, and imaging devices used.

In conclusion, POPF is highly prevalent in pancreatectomy. Current predictive measures to detect POPF intraoperatively have not been successful in decreasing complication rates. ICG perfusion of the pancreatic edge could potentially allow surgeons to predict POPF. We aim to investigate this further with the VIPR-1 trial.

## Supporting information

**S1 File. SPIRIT checklist.**
(PDF)

**S2 File. Administrative approval for the trial.**
(PDF)

**S3 File. Complete study protocol.**
(DOCX)

## Author contributions

**Conceptualization:** Patrick J. Worth.

**Funding acquisition:** Patrick J. Worth.

**Methodology:** Gustavo Salgado-Garza.

**Project administration:** Annika Willy, Flavio G. Rocha.

**Supervision:** Patrick J. Worth.

**Validation:** Patrick J. Worth.

**Visualization:** Brett C. Sheppard.

**Writing – original draft:** Gustavo Salgado-Garza, Annika Willy, Skye C. Mayo, Patrick J. Worth.

**Writing – review & editing:** Flavio G. Rocha, Skye C. Mayo, Brett C. Sheppard, Patrick J. Worth.

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
