## [Decision Letter · Decision Letter 0]

15 Nov 2024

Dear Dr. Worth,

Thank you for submitting your manuscript to PLOS ONE. After careful consideration, we feel that it has merit but does not fully meet PLOS ONE’s publication criteria as it currently stands. Therefore, we invite you to submit a revised version of the manuscript that addresses the points raised during the review process.

We look forward to receiving your revised manuscript.

Kind regards,

Eyüp Serhat Çalık

Academic Editor

PLOS ONE

Journal Requirements: When submitting your revision, we need you to address these additional requirements. 1. Please ensure that your manuscript meets PLOS ONE's style requirements, including those for file naming. The PLOS ONE style templates can be found at https://journals.plos.org/plosone/s/file?id=wjVg/PLOSOne_formatting_sample_main_body.pdf and https://journals.plos.org/plosone/s/file?id=ba62/PLOSOne_formatting_sample_title_authors_affiliations.pdf 2. We note that the original protocol that you have uploaded as a Supporting Information file contains an institutional logo. As this logo is likely copyrighted, we ask that you please remove it from this file and upload an updated version upon resubmission. 3. Thank you for stating the following financial disclosure: "This work is supported by the New Investigator Grant given by the Medical Research Foundation of Oregon." Please state what role the funders took in the study.  If the funders had no role, please state: ""The funders had no role in study design, data collection and analysis, decision to publish, or preparation of the manuscript."" If this statement is not correct you must amend it as needed. Please include this amended Role of Funder statement in your cover letter; we will change the online submission form on your behalf. 4. PLOS requires an ORCID iD for the corresponding author in Editorial Manager on papers submitted after December 6th, 2016. Please ensure that you have an ORCID iD and that it is validated in Editorial Manager. To do this, go to ‘Update my Information’ (in the upper left-hand corner of the main menu), and click on the Fetch/Validate link next to the ORCID field. This will take you to the ORCID site and allow you to create a new iD or authenticate a pre-existing iD in Editorial Manager. 5. Please amend either the title on the online submission form (via Edit Submission) or the title in the manuscript so that they are identical. 6. Please include your full ethics statement in the ‘Methods’ section of your manuscript file. In your statement, please include the full name of the IRB or ethics committee who approved or waived your study, as well as whether or not you obtained informed written or verbal consent. If consent was waived for your study, please include this information in your statement as well. 7. Please review your reference list to ensure that it is complete and correct. If you have cited papers that have been retracted, please include the rationale for doing so in the manuscript text, or remove these references and replace them with relevant current references. Any changes to the reference list should be mentioned in the rebuttal letter that accompanies your revised manuscript. If you need to cite a retracted article, indicate the article’s retracted status in the References list and also include a citation and full reference for the retraction notice.

**Additional Editor Comments:**

Congratulations to the authors for this important study plan. The manuscript was evaluated by three reviewers, their suggestions are given below. Please provide point-by-point answers to the reviewers' questions and make appropriate corrections. We look forward to your uploading the revised version of your manuscript. Successes to you.

Reviewers' comments:

Reviewer's Responses to Questions

**Comments to the Author**

1. Does the manuscript provide a valid rationale for the proposed study, with clearly identified and justified research questions?

Reviewer #1: Yes

Reviewer #2: Yes

Reviewer #3: Yes

2. Is the protocol technically sound and planned in a manner that will lead to a meaningful outcome and allow testing the stated hypotheses?

Reviewer #1: Yes

Reviewer #2: Yes

Reviewer #3: Yes

3. Is the methodology feasible and described in sufficient detail to allow the work to be replicable?

Reviewer #1: Yes

Reviewer #2: Yes

Reviewer #3: Yes

4. Have the authors described where all data underlying the findings will be made available when the study is complete?

Reviewer #1: Yes

Reviewer #2: Yes

Reviewer #3: No

5. Is the manuscript presented in an intelligible fashion and written in standard English?

Reviewer #1: Yes

Reviewer #2: Yes

Reviewer #3: Yes

You may also provide optional suggestions and comments to authors that they might find helpful in planning their study.

Reviewer #1: The submitted protocol is well written in a clear language. I am positive that the study outcome would be interesting to note and might help resolve one significant issue of sequelae of pancreatic surgery.

Reviewer #2: Indocyanine Green (ICG) fluorescence has been use for assessment of tissue perfusion during operative procedures. The topic is nicely selected as POPF is one of the most common complication after pancreatic surgery.

I would like to know from the authors:

1. As trauma is one of the reasons for pancreaticoduodenectomy and authors are going to include all the cases of pancreaticoduodenectomy for this trial, there is no mention of the clinical parameters of patients before surgery and/or any specific set of trauma patients requiring pancreaticoduodenectomy to be included in study.

Trauma patients requiring pancreaticoduodenectomies may be at times clinically stable despite having some degree of global hypoperfusion and/or other associated injuries that can impact the morbidity in such patients. There are some other parameters to categorise such apparently stable patients.

2. What is the scope of blinding only primary surgeon when another surgeon is in operating at that time. It is theoretically ideal to say that the primary surgeon will be out of operating room at the time of ICG administration and assessment, however this reasearch work is going to be a team work and secondary surgeon is most likely to be part of the team. Either blind all of the surgical team or none. Also it is ethically not ideal to send a scrubbed surgeon out of the operating room for the purpose of ICG injection and assessment.

Reviewer #3: I appreciate the opportunity to review the protocol for a phase two trial evaluating the role of intraoperative indocyanine green perfusion of the transected pancreas in predicting postoperative pancreatic leaks. This is a well written protocol stemming from a high volume center for HPB surgery. The authors identify a need for determining intraoperative factors related to POPF, specifically hypoperfusion of the transected neck of the pancreas. They have defined ICG-related measurement criteria a priori, which is critical considering the lack of standard metrics for perfusion. Their methodology is sound for a phase two study. I have minor comments

Introduction

- This can be shortened significantly to 3-4 paragraphs as there are many repetitive themes

- Line 75: ’This study protocol aims to investigate the relationship between hypoperfusion during surgery and the development of postoperative leaks to identify modifiable intraoperative interventions to decrease leak rates.’ Developing modifiable intraoperative interventions does not appear to be the aim of the study. I would recommend rephrasing/removing the second part of this statement.

Outcomes

- Lines 181-186: Define the categories for covariates ’type of pancreatectomy’ and ‘gland texture’. Eg. Would texture refer to hard vs soft as determined intraoperatively?

**Do you want your identity to be public for this peer review?** For information about this choice, including consent withdrawal, please see our Privacy Policy

Reviewer #1: No

Reviewer #2: No

Reviewer #3: **Yes: ** Varun Bansal

---

## [Author Response · Author response to Decision Letter 1]

25 Nov 2024

Reviewer 1:

Dear reviewer, thank you for taking the time to go over our protocol.

1. No revisions suggested.

Reviewer 2:

Thank you for your insights and comments regarding our clinical trial protocol, please find our responses below.

1. We believe that specifying criteria for pancreatectomy after trauma is something that we had left out. Because of this, we now specify in our first inclusion criteria that only elective pancreatectomies are included. These changes can be seen in Page 5, line 106 of the manuscript. We have amended this in our IRB’s entry for the trial protocol as well.

2. The reason behind principal surgeon blinding is to eliminate surgical margin decision intraoperatively based on the metrics captured. The secondary surgeon / first assistant stays scrubbed during the image capturing period of the case. This is done because of institutional guidelines where at least one surgeon needs to stay scrubbed in during the imaging portion of the case. However, all objective metrics (protocol document lines 167-170) are calculated with the imaging platform physically outside of the operating room to avoid impacting any surgical and postoperative treatment decisions. For example, a surgeon or resident might be inclined to pursue a prophylactic measure such as a somatostatin analog to reduce postoperative fistula if they saw that a participant had objective measurements hinting hypoperfusion of the pancreas. Because of this, after the images are captured, the primary surgeon then can re-enter the surgical field and continue with the surgery.

Reviewer 3:

We greatly appreciate your time and efforts while reviewing our manuscript. Please find our point-by-point responses directly below.

1. We have condensed our introduction to 4 paragraphs by deleting redundant sentences. We hope this leads to an easier read while still highlighting the importance of this study. These changes are seen in lines 63-65, 70-74 and 76-79 in the tracked changes document.

2. Line 75 has been edited to remove the statement mentioning a future development of intraoperative interventions because this would be a part of a separate study.

3. Levels for categorical variables for covariates captured are now specified. Type of pancreaticojejunostomy technique, and gland texture – soft, intermediate or firm. (Line 187 and 189).

---

## [Decision Letter · Decision Letter 1]

23 Dec 2024

Dear Dr. Worth,

Thank you for submitting your manuscript to PLOS ONE. After careful consideration, we feel that it has merit but does not fully meet PLOS ONE’s publication criteria as it currently stands. Therefore, we invite you to submit a revised version of the manuscript that addresses the points raised during the review process.

We look forward to receiving your revised manuscript.

Kind regards,

Eyüp Serhat Çalık

Academic Editor

PLOS ONE

Additional Editor Comments:

Dear Authors

Your manuscript after the first revision has been reviewed by the previous reviewers and an additional reviewer and their suggestions are below. We look forward to uploading your manuscript after appropriate revision with your point-by-point answers. All the best.

Reviewers' comments:

Reviewer's Responses to Questions

**Comments to the Author**

1. Does the manuscript provide a valid rationale for the proposed study, with clearly identified and justified research questions?

Reviewer #1: Yes

Reviewer #2: Partly

Reviewer #3: Yes

Reviewer #4: Partly

2. Is the protocol technically sound and planned in a manner that will lead to a meaningful outcome and allow testing the stated hypotheses?

Reviewer #1: Yes

Reviewer #2: Yes

Reviewer #3: Yes

Reviewer #4: No

3. Is the methodology feasible and described in sufficient detail to allow the work to be replicable?

Reviewer #1: Yes

Reviewer #2: Yes

Reviewer #3: Yes

Reviewer #4: No

4. Have the authors described where all data underlying the findings will be made available when the study is complete?

Reviewer #1: Yes

Reviewer #2: Yes

Reviewer #3: Yes

Reviewer #4: Yes

5. Is the manuscript presented in an intelligible fashion and written in standard English?

Reviewer #1: Yes

Reviewer #2: Yes

Reviewer #3: Yes

Reviewer #4: Yes

You may also provide optional suggestions and comments to authors that they might find helpful in planning their study.

Reviewer #1: I appreciate your revisions incorporated as suggested by my worthy reviewer colleague. Its now more appropriate.

Reviewer #2: I am thankful to the authors for their responses.

I would like to have some queries answered by the authors.

1. This protocol is to assess the role of intraoperative ICG perfusion of transected pancreas in predicting leaks and or fistula. however, the primary outcome mentioned is the correlation between ICG measurements and the development or absence of fistula formation. As per International Study Group in Pancreatic Surgery (ISGPS), previously Grade-A fistula is no longer defined as fistula. So, I would like clarity on whether authors are going including biochemical leaks in the study. That has to be clearly mentioned in the title as well as in the text.

2. Oppermann C et al in 2023 noticed that there was a large variation in fluorescence intensity between different organs and between the same organ in different subjects while using a fixed weight-adjusted dosing regimen using the same camera setting and placement(in piglet models). How can this variation be minimized and what would be its effects on the expected outcomes as the hypoperfusion of pancreas would be in relation to other structures in this study.

3. Line 165, "The main outcome will be the development or absence of postoperative pancreatic fistula", however the patients with SBP<90 mmHg, ICU transfer, Blood Transfusion and vasopressor support between postoperative days 3-5 are to be excluded from the study. What should be the standard post-operative time after which a complication could be attributed to pancreatic fistula because all the above-mentioned complications can well happen because of pancreatic fistula. Kindly provide the rationale for excluding these patients from the study.

4. Line-142. "Still pictures will be extracted from the video at 10-second intervals once tissue saturation is stable". For how long will these images be taken?.

5. Line-167 "we will collect surgical drain amylase levels in the first three days after surgery,..) needs to be addressed as POPF as per original ISGPS definition was diagnosed when the amylase content was greater than 3 times the upper normal serum value starting from the postoperative day 3 (rather than first 3 days after surgery). The pancreatic fistula definition per se is unchanged, the criteria for its diagnosis underwent a change.

Reviewer #3: I believe my comments have been addressed appropriately. I would like to applaud the authors again for a well written manuscript.

Reviewer #4: Specific comments:

The study’s key measurements:

Exposure:

ICG measurements of the cut edge of the pancreas will be recorded before reconstruction.

Endpoint:

International Study Group on Pancreatic Surgery criteria for pancreatic fistula will be used to

define leaks and fistulas.

Then the protocol says:

“The primary outcome will be the correlation between ICG measurements and the development or absence of fistula formation.”

This is not the primary outcome, rather it is the study objective: to assess potential association of ICG measurements (X variable, so to speak) and the development or absence of fistula formation (dependent variable Y).

Potential covariates: conventional measurements. What are those?

Sample size calculation needs to be specified:

Anticipating balanced numbers between the two groups, our sample size of 50 participants will have power to detect at least a 25% difference in a leak or fistula rate, assuming a 20% leak rate for the normal ICG perfusion group. We account for a liberal exclusion of 30% of participants due to multiple and complex postoperative factors known to be associated with leaks. Therefore, we anticipate screening 75 participants and enrolling 50.

Do you expect equal number of subjects in the fistula and no fistula groups? Or are you going to control them to be equal?

If so, what’s the difference between them on the ICG measurements?

Or are you going to build a predictive model using ICG measurements to predict the fistula and no fistula status post-surgery as you stated earlier? This requires difference sample size justification.

**Do you want your identity to be public for this peer review?** For information about this choice, including consent withdrawal, please see our Privacy Policy

Reviewer #1: No

Reviewer #2: No

Reviewer #3: **Yes: ** Varun Bansal

Reviewer #4: No

---

## [Author Response · Author response to Decision Letter 2]

4 Feb 2025

Reviewer #1:

I appreciate your revisions incorporated as suggested by my worthy reviewer colleague. It’s now more appropriate.

Response: Reviewer 1, thank you for your valuable feedback.

Reviewer #2:

I am thankful to the authors for their responses.

I would like to have some queries answered by the authors.

1. This protocol is to assess the role of intraoperative ICG perfusion of transected pancreas in predicting leaks and or fistula. however, the primary outcome mentioned is the correlation between ICG measurements and the development or absence of fistula formation. As per International Study Group in Pancreatic Surgery (ISGPS), previously Grade-A fistula is no longer defined as fistula. So, I would like clarity on whether authors are going including biochemical leaks in the study. That has to be clearly mentioned in the title as well as in the text.

Response: Reviewer #2, you raise a very important point regarding the current classification of fistulas and leaks. We intend to evaluate the primary outcome of both fistulas and biochemical leaks in our trial. We have reworded the title and main outcomes to better reflect this. We also intend to conduct a subgroup analysis regarding the predictive potential of ICG measurements for all grade fistulas/leaks, A, B, and C and only clinically relevant, B-C fistulas. The modifications to the manuscript can be seen in lines 1-3, 33, and 176 of the tracked changes document.

2. Oppermann C et al in 2023 noticed that there was a large variation in fluorescence intensity between different organs and between the same organ in different subjects while using a fixed weight-adjusted dosing regimen using the same camera setting and placement (in piglet models). How can this variation be minimized and what would be its effects on the expected outcomes as the hypoperfusion of pancreas would be in relation to other structures in this study.

Response: the works from Opperman et al. were of paramount importance for our study protocol design and we cited it since the first iteration of our submission. Notably, we standardized the camera-organ distance with the built-in function of visible red lasers in the SpyElite platform to diminish variations between one participant and another. These two lasers converge at an optimal distance from the organ and then the measurement is performed. The SpyElite camera hovers above the patient fixed with an arm attached to the main body of the device. This allows for stable and repeatable measurements between participants. In contrast Opperman opted to use the Stryker 1588 which is a hand-held camera, which could be more prone to inter-user variations regarding camera-organ distances. Furthermore, Opperman reports that Tmax and TR–time ratio (T1/2 max/Tmax) appear as the most stable metrics between a piglet model and the next one. We are able to capture these two metrics in our current workflow. Additionally, the camera used by Opperman needs to be switched between visible light and near infrared light, due to the laparoscopic nature of the camera. On the other hand, the SpyElite camera only captures near infrared light when recording. We acknowledge the limitations that could be encountered between participants and hope to report these same variations when we publish our results. We believe that tissue perfusion variation between participants can be accounted for by measuring the difference or delta within the same participant regarding a reference tissue and the target tissue (pancreas). However, the trial itself would explore the impact of these variations to the predictive capabilities of tissue perfusion with ICG. This is further adjusted for with our greater number of planned participants (N = 50) compared to Opperman’s proof of concept study with a porcine model (N = 3).

3. Line 165, "The main outcome will be the development or absence of postoperative pancreatic fistula", however the patients with SBP<90 mmHg, ICU transfer, Blood Transfusion and vasopressor support between postoperative days 3-5 are to be excluded from the study. What should be the standard post-operative time after which a complication could be attributed to pancreatic fistula because all the above-mentioned complications can well happen because of pancreatic fistula. Kindly provide the rationale for excluding these patients from the study.

Response: The reviewer makes an excellent point and our criteria were defined in order to best select out complications from immediate technical errors that could arise postoperatively that might cloud the analysis of pancreatic hypoperfusion intraoperatively (e.g., postoperative bleeding, sepsis, organ failure), or that may subsequently increase the risk of a crPOPF. While it is true that crPOPF can occur in this timeframe and cause these same consequences, 1) this has not been observed in screened patients and 2) critical illness resulting from a crPOPF typically to present with later consequences, which is why we went no further than POD5. Our goal was to attempt to select for leaks that resulted from ischemia detectable intraoperatively and not from other potential causes of hypoperfusion that are not potentially modifiable intraoperatively. It is challenging to define a standard postoperative interval during which a complication could be attributed to a fistula, but as it is common practice to test drain amylase on postoperative days 3-4, we felt excluding immediate postoperative complications after a presumed period of adequate resuscitation would be appropriate.

4. Line-142. "Still pictures will be extracted from the video at 10-second intervals once tissue saturation is stable". For how long will these images be taken?

Response: We have added the minimal duration of capturing ICG metrics with video recording within the SpyElite platform in line 153-154.

5. Line-167 "we will collect surgical drain amylase levels in the first three days after surgery,..) needs to be addressed as POPF as per original ISGPS definition was diagnosed when the amylase content was greater than 3 times the upper normal serum value starting from the postoperative day 3 (rather than first 3 days after surgery). The pancreatic fistula definition per se is unchanged, the criteria for its diagnosis underwent a change.

Response: Thank you for this comment, we have corrected these lines of text in the manuscript to better reflect what is shown in the Table for Schedule of Procedures and Evaluations in the original protocol. Drain amylase is captured at least once during day 3 postop and any subsequent measurements would depend on patient specific scenarios (serial measurements after a high amylase is detected, until drain is removed). These changes can be seen in lines 178-182 from the manuscript with tracked changes enabled.

Reviewer #3:

I believe my comments have been addressed appropriately. I would like to applaud the authors again for a well written manuscript.

Response: Reviewer 3, thank you for your valuable feedback.

Reviewer #4:

The study’s key measurements:

Exposure:

ICG measurements of the cut edge of the pancreas will be recorded before reconstruction.

Endpoint:

International Study Group on Pancreatic Surgery criteria for pancreatic fistula will be used to

define leaks and fistulas.

Then the protocol says:

“The primary outcome will be the correlation between ICG measurements and the development or absence of fistula formation.”

This is not the primary outcome, rather it is the study objective: to assess potential association of ICG measurements (X variable, so to speak) and the development or absence of fistula formation (dependent variable Y).

Response: thank you for your correction. We have made this change, rewording this line to reflect that this is our study objective rather than our primary outcome. These changes can be seen in lines 32-34 of the manuscript with tracked changes.

Potential covariates: conventional measurements. What are those?

Response: we have specified potential covariates and conventional measurements in lines 77-79. Covariates are listed in lines 190-194 of the manuscript with tracked changes.

Sample size calculation needs to be specified:

Anticipating balanced numbers between the two groups, our sample size of 50 participants will have power to detect at least a 25% difference in a leak or fistula rate, assuming a 20% leak rate for the normal ICG perfusion group. We account for a liberal exclusion of 30% of participants due to multiple and complex postoperative factors known to be associated with leaks. Therefore, we anticipate screening 75 participants and enrolling 50.

Do you expect equal number of subjects in the fistula and no fistula groups? Or are you going to control them to be equal?

If so, what’s the difference between them on the ICG measurements?

Or are you going to build a predictive model using ICG measurements to predict the fistula and no fistula status post-surgery as you stated earlier? This requires difference sample size justification.

Response: Imbalance Between Groups:

We do not expect equal numbers of subjects in the fistula and no-fistula groups. Given that approximately 20% of patients in the normal perfusion group are expected to develop a fistula, while 50% of patients in the poor perfusion group are expected to develop one, there will naturally be a greater proportion of non-fistula patients in the normal perfusion group. Based on this, we have calculated the sample sizes for the two groups as follows:

Normal perfusion group (n1): 40 participants

Poor perfusion group (n2): 10 participants

Total sample size: 50 participants

Statistical Justification:

The imbalance ratio between the groups (approximately 4:1) is accounted for in our sample size calculation. The required sample size was determined to achieve 80% power and 5% significance level for detecting a meaningful difference in fistula rates between the groups. We used an effect size calculation based on the difference in expected proportions of fistulas (20% vs. 50%) between the two groups. Changes in the manuscript can be seen in lines 93-103 of the tracked changes file.

Predictive Model:

While we plan to use ICG measurements to predict fistula development, the primary focus of this study is to assess the impact of ICG perfusion on fistula formation post-surgery, with statistical power focused on comparing the proportions of fistulas between the normal and poor perfusion groups. If predictive modeling is pursued in a follow-up analysis, a separate sample size calculation will be performed to determine the appropriate number of participants needed for model training and validation, as this would involve different statistical considerations, including potential adjustments for covariates.

We hope this clarifies our approach to sample size and group imbalance. Please let us know if further elaboration is needed. We hypothesize that ICG can be used in conjunction with other known risk factors to further improve postoperative pancreatic fistula risk prediction.

---

## [Decision Letter · Decision Letter 2]

23 Feb 2025

Dear Dr. Worth,

Thank you for submitting your manuscript to PLOS ONE. After careful consideration, we feel that it has merit but does not fully meet PLOS ONE’s publication criteria as it currently stands. Therefore, we invite you to submit a revised version of the manuscript that addresses the points raised during the review process.

We look forward to receiving your revised manuscript.

Kind regards,

Eyüp Serhat Çalık

Academic Editor

PLOS ONE

Additional Editor Comments:

I would like to thank the authors for their appropriate revisions and point-by-point answers. The manuscript has been re-evaluated by the previous reviewers, and an additional revision proposal is below. We look forward to your re-submission of your manuscript with your responses and appropriate revisions. I wish you success.

Reviewers' comments:

Reviewer's Responses to Questions

**Comments to the Author**

1. Does the manuscript provide a valid rationale for the proposed study, with clearly identified and justified research questions?

Reviewer #2: Yes

Reviewer #4: Partly

2. Is the protocol technically sound and planned in a manner that will lead to a meaningful outcome and allow testing the stated hypotheses?

Reviewer #2: Yes

Reviewer #4: No

3. Is the methodology feasible and described in sufficient detail to allow the work to be replicable?

Reviewer #2: Yes

Reviewer #4: No

4. Have the authors described where all data underlying the findings will be made available when the study is complete?

Reviewer #2: Yes

Reviewer #4: Yes

5. Is the manuscript presented in an intelligible fashion and written in standard English?

Reviewer #2: Yes

Reviewer #4: Yes

You may also provide optional suggestions and comments to authors that they might find helpful in planning their study.

Reviewer #2: Thank you for handling the queries and updating the manuscript. The manuscript is more appropriate now.

I suggest minor polishing of lines 153-154 of tracked changes document.

Reviewer #4: The protocol states:

“We anticipate a 20% incidence of fistula in the normal perfusion group, and at least a 50% incidence in the poor perfusion group is expected. With an expected imbalance ratio of approximately 4:1 between these groups, a total sample size of 50 participants (40 in the normal perfusion group and 10 in the poor perfusion group) provides 80% power at a 5% significance level to detect this difference.”

My calculations show that:

40 in the normal perfusion group and 10 in the poor perfusion group with expected 20% and 50% incidence of fistula, the study has 49.1% power using chi-squared test and 35% power using exact test.

Could you explain how did you get 80% power?

**Do you want your identity to be public for this peer review?** For information about this choice, including consent withdrawal, please see our Privacy Policy

Reviewer #2: **Yes: ** Irshad Ahmad

Reviewer #4: No

---

## [Author Response · Author response to Decision Letter 3]

8 Apr 2025

Reviewer #2:

Thank you for handling the queries and updating the manuscript. The manuscript is more appropriate now.

I suggest minor polishing of lines 153-154 of tracked changes document.

Response from authors: Dear Reviewer, we have now improved the wording of the sentence you pointed out. Thank you for your comments. These changes are in the tracked manuscript and in the latest revised version.

Reviewer #3:

The protocol states:

“We anticipate a 20% incidence of fistula in the normal perfusion group, and at least a 50% incidence in the poor perfusion group is expected. With an expected imbalance ratio of approximately 4:1 between these groups, a total sample size of 50 participants (40 in the normal perfusion group and 10 in the poor perfusion group) provides 80% power at a 5% significance level to detect this difference.”

My calculations show that:

40 in the normal perfusion group and 10 in the poor perfusion group with expected 20% and 50% incidence of fistula, the study has 49.1% power using chi-squared test and 35% power using exact test.

Could you explain how did you get 80% power?

Response from authors:

Dear Reviewer,

Thank you for your meticulous review and for identifying the inconsistency in our power calculation. We sincerely appreciate your diligence. Below, we clarify the steps taken to address this:

The manuscript initially incorrectly stated a 50% fistula incidence in the poor perfusion group. This was a typographical error; our original calculations (and the cited reference) used 65%, derived from Chen et al. (Ref 31), who reported a 67% incidence of crPOPF in hypoperfused patients. We have corrected this value in the revised manuscript.

Our sample size calculation assumes a two-sample proportion test with unequal group sizes, using the normal approximation method. While exact tests (e.g., Fisher’s) often require larger samples, this approach is widely accepted for study design when preliminary effect sizes are based on prior data. The formula accounts for:

o Expected proportions (20% vs. 65%)

o A 4:1 enrollment ratio (clinically justified by the higher prevalence of normal perfusion cases)

o 80% power and a two-sided α = 0.05.

Validation of Calculations:

Using the script below (executed in R 4.3.3), we calculated a required 53 participants (42 normal perfusion, 11 poor perfusion). This aligns with the “pwr.2p2n.test” function using the “pwr” package when accounting for unequal group sizes, yielding 80.3% power. Our calculation provides a conservative yet feasible estimate for protocol planning.

---

## [Decision Letter · Decision Letter 3]

29 Apr 2025

The VIPR-1 trial (Visualizing Ischemia in the Pancreatic Remnant) - Assessing the role of intraoperative indocyanine green perfusion in predicting postoperative pancreatic leaks and fistulas: protocol for a phase II clinical trial.

PONE-D-24-39874R3

Dear Dr. Worth,

We’re pleased to inform you that your manuscript has been judged scientifically suitable for publication and will be formally accepted for publication once it meets all outstanding technical requirements.

Kind regards,

Eyüp Serhat Çalık

Academic Editor

PLOS ONE

Additional Editor Comments (optional):

Reviewers' comments:

Reviewer's Responses to Questions

**Comments to the Author**

1. Does the manuscript provide a valid rationale for the proposed study, with clearly identified and justified research questions?

Reviewer #4: Partly

2. Is the protocol technically sound and planned in a manner that will lead to a meaningful outcome and allow testing the stated hypotheses?

Reviewer #4: No

3. Is the methodology feasible and described in sufficient detail to allow the work to be replicable?

Reviewer #4: No

4. Have the authors described where all data underlying the findings will be made available when the study is complete?

Reviewer #4: No

5. Is the manuscript presented in an intelligible fashion and written in standard English?

Reviewer #4: Yes

You may also provide optional suggestions and comments to authors that they might find helpful in planning their study.

Reviewer #4: For a study of this nature, the study team seems to rely on journal reviewers to get the statistical considerations right or merely consistent, i.e. changing numbers to address reviewer’s critique. This is a serious erosion of the confidence that the study team has the necessary expertise to design and execute the study.

**Do you want your identity to be public for this peer review?** For information about this choice, including consent withdrawal, please see our Privacy Policy

Reviewer #4: No

---

## [Editor Report · Acceptance letter]

PONE-D-24-39874R3

PLOS ONE

Dear Dr. Worth,

I'm pleased to inform you that your manuscript has been deemed suitable for publication in PLOS ONE. Congratulations! Your manuscript is now being handed over to our production team.

Kind regards,

on behalf of

Dr. Eyüp Serhat Çalık

Academic Editor

PLOS ONE